# Human Pluripotent Stem Cell-Based Models for Hirschsprung Disease: From 2-D Cell to 3-D Organoid Model

**DOI:** 10.3390/cells11213428

**Published:** 2022-10-29

**Authors:** Kathy Nga-Chu Lui, Elly Sau-Wai NGAN

**Affiliations:** Department of Surgery, Li Ka Shing Faculty of Medicine, The University of Hong Kong, Pokfulam, Hong Kong

**Keywords:** Hirschsprung disease, induced pluripotent stem cells, disease modeling, enteric nervous system, colonic organoids

## Abstract

Hirschsprung disease (HSCR) is a complex congenital disorder caused by defects in the development of the enteric nervous system (ENS). It is attributed to failures of the enteric neural crest stem cells (ENCCs) to proliferate, differentiate and/or migrate, leading to the absence of enteric neurons in the distal colon, resulting in colonic motility dysfunction. Due to the oligogenic nature of the disease, some HSCR conditions could not be phenocopied in animal models. Building the patient-based disease model using human induced pluripotent stem cells (hPSC) has opened up a new opportunity to untangle the unknowns of the disease. The expanding armamentarium of hPSC-based therapies provides needed new tools for developing cell-replacement therapy for HSCR. Here we summarize the recent studies of hPSC-based models of ENS in 2-D and 3-D culture systems. These studies have highlighted how hPSC-based models complement the population-based genetic screens and bioinformatic approaches for the discovery of new HSCR susceptibility genes and provide a human model for the close-to-physiological functional studies. We will also discuss the potential applications of these hPSC-based models in translational medicines and their advantages and limitations. The use of these hPSC-based models for drug discovery or cell replacement therapy likely leads to new treatment strategies for HSCR in the future. Further improvements in incorporating hPSC-based models with the human-mouse chimera model and organ-on-a-chip system for establishing a better disease model of HSCR and for drug discovery will further propel us to success in the development of an efficacious treatment for HSCR.

## 1. Introduction

Hirschsprung disease (HSCR), which is also known as congenital aganglionic megacolon, is a neurocristopathy caused by the incomplete colonization of the colon by enteric neural crest cells (ENCCs). HSCR is a multigenic congenital disorder with over 80% heritability [1], and its incidence is approximately 1 per 5000 births worldwide, with the highest incidence rate (1.4 per 5000 births) in the Asian population. The incidence rate in males is 3–4 times higher than in females [2,3]. Missing enteric nervous system (ENS) in the colon causes uncoordinated muscular peristalsis and functional bowel obstruction. As a consequence, HSCR patients experience delayed passage of meconium in the first two days after birth and suffer from vomiting, feeding difficulties, abdominal distension, and constipation [4].

HSCR patients are classified into three subtypes based on the extent of aganglionosis (i.e., bowel segment without enteric neurons), namely short-segment HSCR (S-HSCR), long-segment HSCR (L-HSCR) and total colonic aganglionosis (TCA). S-HSCR accounts for around 80% of the cases, while 15% and 5% of the HSCR cases are L-HSCR and TCA, respectively. Different genetic architectures are found in these HSCR subtypes [5]. L-HSCR and TCA mostly are autosomal dominant, while the S-HSCR follows non-Mendelian oligogenic inheritance where patients may carry different constellations of coding and noncoding variants leading to the malformation of the ENS. The disease risk and expressivity are influenced by the underlying genetic background of the individuals [6]. The oligogenic nature of HSCR makes it challenging to build a definitive diagnostic framework for prenatal genetic screening. Till now, surgical removal of the aganglionic segment of the colon is still the only available treatment for HSCR. However, many patients still suffer from constipation and enterocolitis after surgery, particularly short bowel syndrome frequently happens in TCA patients [4]. Therefore, tremendous efforts have been made to develop a new and efficacious treatment for HSCR patients.

Genetic variants or mutations in *RET*, *ZEB2*, *EDNRB*, *SOX10*, *L1CAM*, *PHOX2B*, *GDNF*, *NRTN, EDN3, ECE1, GFRA1, NRG1* were identified in HSCR patients, imparting up to 50% of the cases [3,7,8,9]. These disease-associated genes encode signaling molecules, cell surface receptors, trophic factors, or transcription factors which are essential for the proper development of the ENS (Figure 1). The individual effects of these disease-associated genes have been studied using the genetic approach through the generation of mutant animal models of the corresponding genes [10]. Currently, the mouse, rat, chick, *Xenopus* and zebrafish are still the most popular animal models used for studying HSCR and other congenital disorders. Compelling evidence suggests that genetic background influences the disease expressivity of HSCR, especially the epistasis among various coding and noncoding variants, a patient-based model which carries the matched genetic background represents an alternative tool for bridging the gaps between human genetic screens and functional studies performed with animal models. In particular, current advances in stem cell research enable the use of human induced pluripotent stem cells (hPSCs) derived from the patient’s somatic cells to generate two-dimensional (2-D) ENS models and even three-dimensional (3-D) “mini-gut” to recapitulate the human HSCR conditions. In this review, we will summarize the recent findings from the studies of hPSC-based models of HSCR and highlight their potential applications in drug discovery and cell replacement therapy.

## 2. hPSC-Based Disease Models of HSCR

### 2.1. In Vitro 2-D ENS Model

By mimicking the endogenous signaling events that happened during the neural crest cell (NCC) formation, human pluripotent stem cells (hPSCs) can form NCCs with sequential activation and inhibition of various developmental cues. To generate ENCCs from hPSC, hPSCs are first directed to the ectoderm lineage using dual SMAD inhibition where both BMP and TGF-β signaling pathways are inhibited by the addition of BMP antagonist (LDN193189) and TGF-β antagonist (SB431542). Subsequent activation of WNT signaling using WNT agonist CHIR99021 further guides the ectoderm cells to NCC lineage [11]. The NCCs are then caudalized to vagal lineage by the treatment with retinoic acid, resembling the in vivo somitic environment. The vagal NCC-like cells can then be enriched by cell sorting using ENCC-specific markers, such as p75^NTR^, HNK1, and RET. Various types of enteric neurons can be derived from these ENCC-like cells by promoting their neuronal lineage differentiation by adding different neurotrophic factors [11,12,13].

Using this stepwise induction protocol, hPSCs derived from healthy individuals or HSCR patients can generate the disease-relevant cells (i.e., ENCCs and enteric neurons) in a progressive differentiation manner resembling the in vivo developmental processes of the ENS. It is noteworthy that cells derived from patient hPSC lines carrying exactly the same genetic makeup as the HSCR patient, the global and accumulative effects of various genetic mutations or variants on the ENS development can be recapitulated in vitro. The specific developmental process interfered with by the mutations can be further elucidated based on the changes in their cellular phenotypes compared to the isogenic control or the corresponding control group.

With CRISPR-Cas9 genome editing techniques [14,15,16,17], a single or multiple specific HSCR-associated variants or mutations can be introduced into the genome of healthy (control) hPSCs to generate “diseased” ENCCs. Similarly, the disease-associated variant(s) or mutation(s) can be “corrected” in the patient-derived hPSCs to illustrate the primary biological implications of the particular variant(s) or mutation(s) [6,18,19]. A proof-of-concept study was conducted to demonstrate the dose-dependent effect of *RET* on the differentiation and migration of hPSC-derived ENCCs. A heterozygous or homozygous deletion of *RET* was introduced into the control hPSC line. ENCCs derived from these mutant hPSC lines exhibited severe defects in making enteric neurons and failed to migrate, as monitored by the in vitro differentiation and stretch assays. In line with these observations, correcting a deletion mutation in *RET* from a TCA-hPSC line could nicely restore the functions of ENCCs [19]. Moreover, by cross-referencing the transcriptome profiles of the patient-derived ENCCs with the whole exome sequencing data of the corresponding patient, a novel mutation in vinculin has been identified, and the subsequent rescue experiment with a “corrected” hPSC line directly illustrated the functional impact of vinculin in ENCC development [19].

With the rapid development of the whole genome genetic screens, hundreds of genetic variations have been identified in HSCR patients. hPSC-based model of HSCR has been used to complement the whole genome genetic screen to unbiasedly discover the causative mutations, with defined disease mechanisms [6,18]. For instance, the analysis of transcriptomic profiles of HSCR-ENCCs from different subtypes of HSCR revealed that the common and distinctive biological pathways are dysregulated in HSCR patients with different *RET*-sensitized genetic backgrounds. More importantly, the biological implications of a novel HSCR susceptibility gene, *BACE2*, which was identified from a high coverage whole-genome sequencing of S-HSCR, could be demonstrated using an hPSC-based model of the ENS and a new disease mechanism involving the BACE1-APP-BACE2 pathway underlying the HSCR pathogenesis has been proposed [6].

Unveiling the noncoding regions of the human genome, particularly defining the disease-relevant functional noncoding variants, remains the most formidable challenge in the field of human genetics. To tackle this issue, the epigenome of hPSC-derived ENCCs was obtained to define the ENS-specific noncoding regulatory regions. With the support of a novel association and prioritization bioinformatic framework that considers convergent effects of different genetic variants in one or more regulatory regions of the same gene and uses transcription factor binding motifs as a functional proxy, additional novel HSCR susceptibility loci have been identified. Functional assays in hPSCs-derived ENCCs further confirmed the regulatory role and HSCR-relevance of an enhancer of Phosphatidylinositol-4-phosphate 3-kinase C2 domain-containing beta polypeptide (PIK3C2B) and a long-range enhancer in *RET* [18]. This study has clearly demonstrated the power of integrative analysis, where integration of the stem cell and bioinformatic platforms allows us to have a more sophisticated cause-and-effect study of the disease-relevant variants or mutations that facilitates the reconstruction of an HSCR-related regulatory network and to better define the molecular mechanisms underpinning the HSCR pathogenesis.

### 2.2. 3-D Human Colonic Organoids

Continuous interactions between the ENCCs with the endodermal epithelium and the adjacent gut mesenchyme are essential for the proper development of the ENS, and disruption in these interactions may lead to HSCR. It is conceivable that a three-dimensional (3-D) “mini-gut” composed of the intestinal epithelium and ENCCs derived from the patient hPSCs may provide a close-to-physiological environment and, thus, a better disease model of HSCR. By resembling the embryonic intestinal development with a series of growth factor manipulations, hPSCs can be induced into the endodermal lineage to generate intestinal epithelium, which can later self-organize into a 3-D organ-like structure called intestinal organoids (HIOs) [20,21]. By incorporating hPSC-derived ENCCs into HIOs, a “mini-gut” with the intestinal epithelium and the ENS can be generated. This “mini-gut” can then be engrafted into mouse kidney capsules to develop smooth muscle layers in vivo and thereby become a more mature intestinal tissue with a similar cytoarchitecture as seen in the human intestine. The enteric neurons in the “mini-gut” can also respond to electrical stimulation and trigger the contractile movement in the muscle wall of the “mini-gut” [22,23,24]. With a brief activation of the BMP signaling, the HIOs can be directed to a more posterior fate to generate colonic organoids (HCOs) [24,25,26]. Since the HSCR defects mostly occur in the colon regions, the HCOs can serve as a more relevant disease model for HSCR.

The first attempt to establish an organoid model of enteric neuropathy was reported by Workman et al. using hPSC-derived HIOs in which both wild-type and *PHOX2B* mutant (*PHOX2B^Y14X/Y14X^*) ENCCs were recombined with wild-type HIOs. The authors demonstrated the cell-autonomous effect of PHOX2B on the development of ENS in this ex vivo organoid model [22]. More intriguingly, this study also revealed the ENCCs-mesenchyme interaction and showed that ENCCs have non-cell-autonomous effects on the development of smooth muscle cells in HIOs [22]. Similar observations were reported in a more recent study from the same group. The authors showed that ENCCs promote the development of gut mesenchyme and the glandular morphogenesis of antral stomach organoids engineered from three primary germ layers derived separately from hPSCs [27]. These studies highlight the potential use of an organoid model for exploring the potential interactions between ENCCs and their neighboring cells. Following the same idea, the ENCCs derived from “diseased” hPSCs can be incorporated with the colonic organoids derived from the control or “diseased” hPSCs to generate human “diseased” colon-like structures for evaluating the biological impacts of the HSCR-associated mutations, in which the environmental effects can also be taken into considerations. Moreover, the functionality of the diseased enteric neurons and the environmental effects can be assessed through the contractile motions of the “diseased” “mini-gut” to unveil how the HSCR mutations are correlated to neuromuscular transmission.

An overview of 2-D and 3-D hPSC-based ENS models in HSCR-related studies is shown in Figure 2.

## 3. 2-D ENS Cell Model vs. 3-D Organoid Model: Advantages and Limitations

Different in vitro models for various congenital disorders could be derived from hPSC through the stepwise manipulations of different signaling molecules and growth factors. Through the 2-D hPSC-derived cell model of ENS, the neuronal differentiation trajectory during hPSC-to-ENCC-to-enteric neuron transitions can be monitored at both cellular and transcriptomic levels [24]. The 2-D model allows easy assessment of the differentiation competency and the specification of specific subtypes of enteric neurons through immunostaining with different cell markers. However, since the ENCC derivatives in the 2-D culture grow randomly in an unorganized manner, different subtypes of ENCC derivatives are incapable of forming specific cell-cell interactions. As a result, how the ENCC derivatives organize into networks of enteric ganglia cannot be addressed with the 2-D model of ENS. In addition, the 2-D ENS model only contains cells of a single lineage which does not allow the investigation of the interplay between the ENS and other germ layers like the intestinal epithelium and smooth muscle.

Due to the above limitations of the 2-D culture model, the 3-D organoid model nowadays becomes an increasingly popular option for recapitulating human in vivo developmental processes [28]. In the HCOs, the 2-D ENS cells can be integrated with the gut epithelium spheroids to generate innervated HCOs which can then be engrafted in the mouse kidney capsule for maturation to become a “mini-gut” with smooth muscle layers. The three germ layers of cells in the “mini-gut” were well-organized with cytoarchitecture resembling the in vivo gut, in which the colonic epithelia with defined crypts were surrounded by layers of smooth muscle fibers aligned with enteric neurons and glia. In the 3-D organoids, cells at different cell lineages were co-cultured together. The growth factors secreted from the cells of endoderm origin can support the maturation of the ENS, such that more subtypes of neural cells, as well as glial cells, can be found in the “mini-gut.” Therefore, the 3-D organoids allow the study of the interplay between the enteric neurons and the gut mesenchyme during ENS development. However, the generation of “mini-gut” is much more complicated, time-consuming, and technically challenging. There are significant variations among the engrafted colonic organoids after the long-term maturation in the murine kidney capsule in vivo.

The advantages of the 2-D and 3-D hPSC-based disease models of ENS are summarized in Figure 3.

## 4. hPSC-Based HSCR Disease Model for Drug Discovery

Besides studying the disease etiologies of HSCR, ENS cells derived from the HSCR-hPSC lines can be used for drug screening to discover novel treatments that can potentially cure the disease. For example, HSCR-ENCCs derived from *EDNRB^−/−^* human embryonic stem cells (hESCs), which showed migration defect in vitro and in vivo after transplantation to mouse gut, were used as a disease model for screening of the small molecules for HSCR treatment. Through the drug screening in the 2-D ENS model of HSCR, pepstatin A was identified to rescue the migration defect in the *EDNRB^−/−^* ENCCs in a dose-dependent manner, implying that pepstatin A could be a potential drug for improving the colonization of ENCCs in the colon [13].

While the in vitro 2-D ENS model can be a tool for evaluating the migration and differentiation capabilities during drug screening, the ability of the treated HSCR-ENCCs to trigger muscle contraction cannot be assessed. With the 3-D “mini-gut” model, HSCR-ENCCs derived from patient hPSCs can be incorporated into the muscle layer of the HCOs. The drug effects can be monitored based on how it improves the functionality of the patient enteric neurons in inducing contractile motions of the “mini-gut.” Nevertheless, the current protocol for the generation of “mini-gut” requires engraftment to the mouse kidney capsule for tissue maturation, which makes it incompatible with the high-content drug screen. Thus, the “mini-gut” model can be considered an additional platform for the tertiary drug screen for refining the drug potencies or establishing personalized medicine. In other words, the 2-D and 3D models can be seen as complementary.

## 5. hPSC-Based Replacement Therapy for HSCR

The cause of HSCR is the absence of ENCCs in the colon, so transplantation of ENCCs to the aganglionic colonic segment should help restore the muscle contractibility of the bowel and rescue the functional defects. Multiple studies have shown that ENCCs derived from different cell sources, such as endogenous ENCCs, neural stem cells (NSCs), and hESCs, can be transplanted to the aganglionic colon and functionally integrate with the endogenous ENS to improve bowel motility [13,29,30,31,32,33,34,35,36,37,38,39,40,41,42,43,44,45]. Nevertheless, there are a considerable number of difficulties that need to be overcome before using these sources of stem cells for replacement therapy for HSCR. For instance, obtaining enough endogenous ENCCs or autologous postnatal NSCs from the patients for autogenous transplantation is challenging. In addition, using hESCs as regenerative medicine remains ethically controversial as they are obtained from human embryos. The non-autologous source of hESCs also poses a risk of an undesired immune response, such as graft rejection upon transplantation [46]. To address these technical, ethical, and safety issues, hPSC-derived ENCCs and/or enteric neurons would be a solution.

To minimize the post-transplantation graft rejection, hPSCs can be reprogrammed from autologous somatic cells obtained from the skin biopsies of the HSCR patients by overexpressing the four reprogramming factors (Oct3/4, Klf4, Sox2, and c-Myc) [47]. By correcting the disease-associated variants or mutation(s) using seamless genome-editing techniques such as the CRISPR-Cas9 system [15,17], the hPSCs derived from HSCR patients can be used to generate “healthy” ENCCs for replacement therapy for HSCR. Previously, our group demonstrated that the migration and neuronal differentiation defects of the HSCR-hPSC-derived ENCCs could be rescued by correcting the inherited disease-associated mutations with the CRISPR-Cas9 system [19]. Although this study has highlighted the feasibility of using patient hPSC for the generation of functional autologous ENCCs for replacement therapy, this approach would be applicable to patients with a defined mutation in a single gene. Using human leukocyte antigen (HLA)-matched hPSC for transplantation would be an alternative way to minimize immune rejection. Instead of preparing hPSC lines with thousands of unique HLA haplotypes, Taylor et al. have proposed to establish a panel of ten hPSC lines homozygous for common HLA types from 10,000 donors to provide a complete HLA-A, HLA-B, and HLA-DR match for 37.7% of recipients and a beneficial match for 67.4% in the population [48]. More recently, the HLA cloaking approach has emerged with the advent of CRISPR gene editing technologies to generate the universal hPSC lines for transplantation, where immunogenicity-associated HLA genes are inactivated in hPSC to enhance immune compatibility [49,50].

While direct transplantation of ENCCs is a promising approach for cell replacement therapy for HSCR, the emergence of colonic organoid technology may allow the creation of an artificial colon for engraftment into the patient, representing an alternative treatment for HSCR. Different cell lineages within the 3-D organoids can self-organize to produce cytoarchitecture resembling the in vivo tissues, so it should be feasible to use organoids to build an artificial organ for replacement therapy [51,52,53,54,55,56,57,58,59]. It has been demonstrated that the transplanted colonic organoids can grow and be maintained for a long period of time in vivo after engrafting into the colon epithelium [60], suggesting that the ENS-innervated colonic organoids also have the potential to functionally integrate with the endogenous colon of the HSCR patients. However, using hPSC-derived colonic organoids for HSCR treatment is still a preliminary idea, as the maturation of ENS-innervated colonic organoids needs to be taken place inside the body, and it remains challenging to regulate the whole developmental process. It may also be necessary to culture the colonic organoid in synthetic scaffolds so they can grow in a more organized manner. Recently, researchers have attempted to incorporate blood vessels into the hindgut organoids to generate vascularized colonic organoids in vitro, but they failed to integrate the ENS cells into these vascularized colonic organoids [61]. This indicates that vascularization of the transplanted colonic organoids in situ would be another issue needed to be addressed. Other concerns, such as large-scale cell culture in Good Manufacturing Practice (GMP) grade, long-term survival, tumorigenicity, immunogenicity, and functional integration between the transplanted tissues and the endogenous tissues, must also be addressed before considering using the organoids for HSCR treatment.

## 6. Future Perspectives of the hPSC-Based Disease Models of HSCR

### 6.1. Human-Mouse Chimera Disease Model

Although progressive improvements have been made in the generation of the colonic organoid model from hPSCs, there are still considerable existing problems, such as the formation of tubular structures resembling the real gut. Human-mouse chimera can therefore be established and used as an in vivo disease model carrying human characteristics. A previous study has revealed that the migration of hPSC-derived NCCs follows the same path as the endogenous murine NCCs upon injection into the mouse gastrula. Most intriguingly, the engrafted human NCCs can be successfully committed to the melanocyte lineage and subsequently differentiate into mature pigment cells residing in the skin of the human-mouse chimera [62]. This experimental approach allows almost the entire developmental process of a specific hPSC-derived NCC lineage to take place in the host animals, so all the human NCCs can develop fully under physiological conditions. Later, the same group of scientists used this experimental paradigm to generate a human-mouse chimera of neuroblastoma derived from hPSC-derived NCC to monitor the tumor initiation, progression, manifestation, and tumor-immune-system interactions in vivo [63]. This demonstrates this experimental approach’s feasibility in generating human-mouse chimera for modeling the development of different NCC lineages. These studies provide the foundation for generating humanized mouse models of HSCR with “diseased” hPSCs-derived ENCCs to support the in vivo study of human-specific HSCR mutations.

### 6.2. Organ-On-A-Chip

To produce well-organized complex organoids that incorporate more different types of tissues and environmental factors, researchers have started using an organ-on-a-chip system to generate 3-D organoids. Intestinal epithelial cells cultured in a microfluidic gut-on-a-chip device can spontaneously undergo 3-D morphogenesis and organize into crypt-villus microstructures [64,65,66,67]. The porous membrane on the microfluidic chip allows the exchange of nutrients and oxygen in the cultured tissues, similar to the vascularization system inside the body. Gut microbiota can also be introduced into the intestine “lumen” within the gut-on-a-chip device to study the gut-microbiota interactions. It is known that the microbiota within the gut influences the development of the central nervous system via the microbiota-gut-brain axis, so there are also increasing views that the gut microbiota can affect the development of ENS [68,69,70]. Since it has been demonstrated that hPSC-derived neural stem cells can co-culture with endothelial cells to study blood-brain-barrier functions using the organ-on-a-chip system [71], there is a high probability of co-culturing hPSC-derived ENCCs with the epithelial cells as well as muscle cells in a gut-on-a-chip after further advancement in the design of the microfluidic device and optimization of the culture system. In addition, intestinal organoids have been proposed to be used as an in vitro model of necrotizing enterocolitis (NEC), a gut inflammatory disorder, where stress factors, such as hypoxia and LPS, or various cytokines (TNF-**α** and IFN-**γ**) were used to induce intestinal epithelial injury or the inflammatory conditions in an organoid model [72]. In line with this idea, a clinical condition, namely HSCR-associated enterocolitis (HAEC), can be recapitulated on the chips. In particular, the gut microbiota-induced intestinal epithelial injury or the inflammatory conditions could be mimicked using various stress factors or cytokines. Then, inflammatory responses under the HSCR conditions can also be investigated. More importantly, the patient’s drug response can be assessed with the system for the development of personalized medicine.

A summary of these two future research directions is shown in Figure 4.

## 7. Conclusions

Both the 2-D ENS culture model and the 3-D colonic organoid model derived from hPSCs are suitable to be used as disease models for HSCR. Each model has its own advantages and limitations when it is used to recapitulate the in vivo developmental processes. In most cases, both 2-D and 3-D culture models would be used in complement with each other to unveil the molecular mechanisms underlying disease pathogenesis thoroughly. In addition, both 2-D and 3-D models of ENS can be utilized as a drug screening platform to discover novel molecules which can improve ENS development or gut motility. With the rapid evolution of stem technologies and the tools for cell-based treatments, we envision that a standard protocol for the generation of GMP-grade ENCCs will be established with the HLA-compatible hPSC lines and used as regenerative medicine for HSCR patients in the future.

## Figures and Tables

**Figure 1 cells-11-03428-f001:**
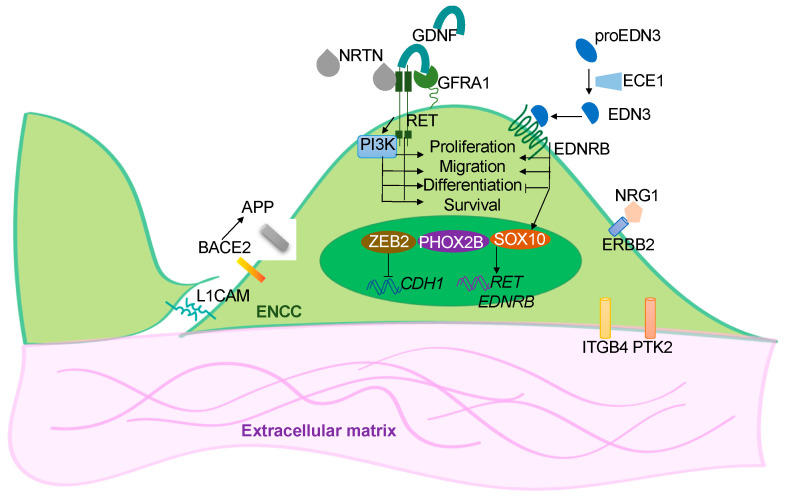
Overview of the functions of the disease genes. Simplified illustration showing the functional roles of the HSCR-associated genes in ENS development. These disease-associated genes encode for the cell surface receptors (RET, EDNRB and ERBB2) in the ENCCs and their ligands (GDNF, NRTN, EDN3, ECE1 and NRG1); the transcriptional factors (SOX10, ZEB2 and PHOX2B); or the cell-cell (L1CAM) and cell-extracellular matrix (IGGB4, PTK2) interacting molecules. BACE2 is a protease that acts on APP to prevent the production of toxic Ab peptides from protecting the survival of ENCCs.

**Figure 2 cells-11-03428-f002:**
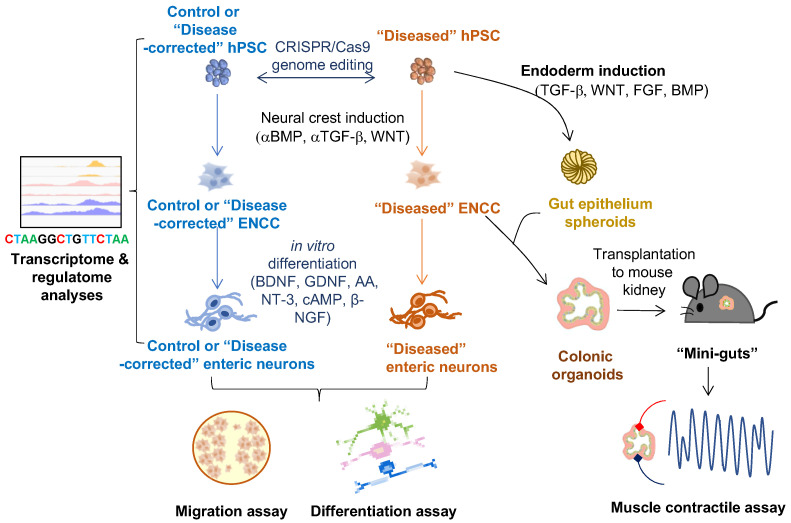
Applications of hPSC-derived 2-D and 3-D cell models of ENS on studying HSCR disease. ENCC-like cells and intestinal organoids are derived from hPSCs by manipulating different developmental cues (BMP, TGF-b, WNT, and FGF). Innervated “mini-gut” are generated by combining the intestinal organoids with the ENCCs and engrafted to the mouse kidney capsule for further maturation. The 2-D cell models (ENCCs and enteric neurons) can be used for multi-omics studies (e.g., transcriptome and regulatome analyses) and functional assays (e.g., migration and in vitro differentiation assays). The 3-D cell models (colonic organoids and “mini-guts”) can be used for neuromuscular studies such as muscle contractile assay.

**Figure 3 cells-11-03428-f003:**
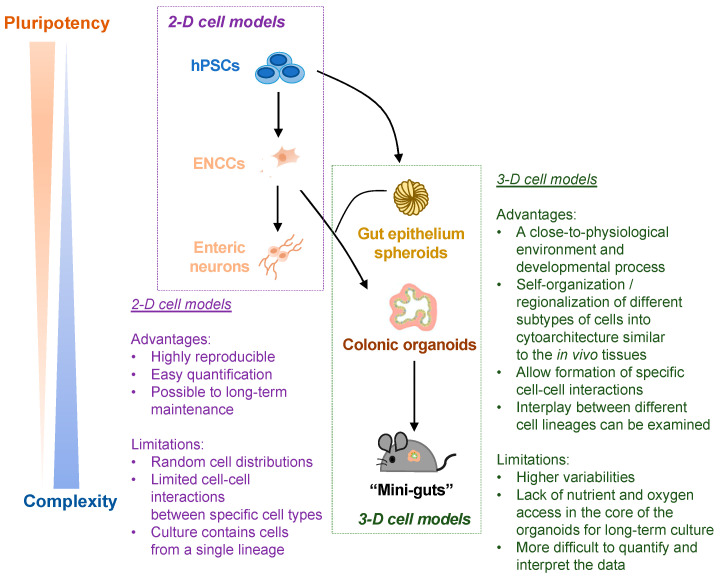
Schematic summary of the advantages and limitations of hPSC-derived 2-D and 3-D cell models of ENS. Both 2-D cell models (ENCCs and enteric neurons) and 3-D cell models (colonic organoids and “mini-guts”) are used for HSCR-related studies. The advantages and limitations of 2-D vs. 3-D cell models are listed.

**Figure 4 cells-11-03428-f004:**
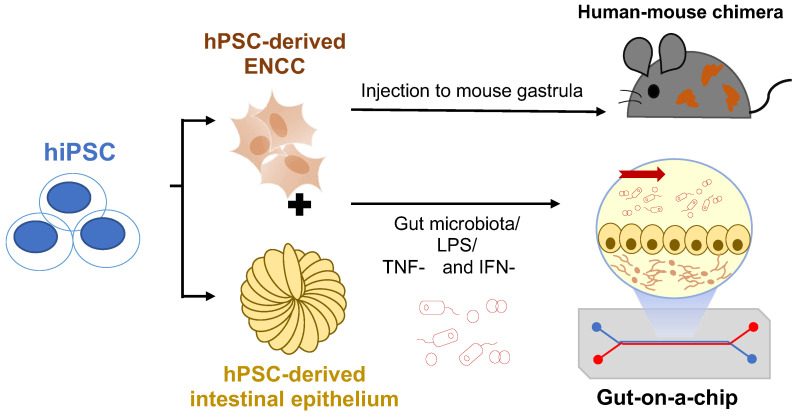
Future perspectives of the hPSC-based disease models. hPSC-derived NCCs can be injected into mouse gastrula to generate human-mouse chimera for in vivo disease models. Co-culture of ENCCs and intestinal epithelium with a gut-on-a-chip device allows the study of microbiota-gut-ENS interactions in HSCR. LPS: Lipopolysaccharides; TNF-α:Tumor necrosis factor-alpha; IFN-γ: Interferon-gamma.

## Data Availability

Not applicable.

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
