# Peer review of "Human Pluripotent Stem Cell-Based Models for Hirschsprung Disease: From 2-D Cell to 3-D Organoid Model"

_cells, 2022, doi:10.3390/cells11213428_

Round 1

Reviewer 1 Report

Thank you for having me review this excellent review article. This is a well-written manuscript with extensive literature review. A topic that the authors selected was critically important and advanced dramatically in this research field for the last 5-10 years. Papers that the authors cited are all updated and relevant. A few minor grammar errors were seen as listed below and a few minor revisions below would be worth considering for improving this article before publication.

Minor points:

1. In page 2, Figure1. The letters written in the figure are very small and difficult to read. Particularly, the colors of the letters written in the muscle layer (saying, "Extracellular matrix"?) is the same one as the background and it's extremely difficult to read. Please consider using a different color that gives more contrast to the background.

2. In page 5, line192, "disease enteric neurons" should be "diseased"??

3. In page 7, line 282, "our group has showed" should be "shown".

4. In page 8, line355-357, The authors described the usefulness of "Organ-on-a-chip" system for studying the HSCR-associated enterocolitis (HAEC), however, this ex vivo cell culture system wouldn't contain immune cells. It is not clear how useful this devise can be when studying an inflammatory condition without involving the immune cells. The authors should discuss this perspective a little more.

5. "the" is usually required when describing the "ENS". Please double check the entire manuscript.

6. In page 3, lines 90-91, please check the font of TGF-"b" 

Reviewer 2 Report

This Review addressed a very intriguing issue namely stem cells potential for treatment of intestinal aganglionosis. I have some concerns, mostly referred to the introduction section

1) It is incorrect to state: "HSCR is a multigenic congenital disorder with over 80% heritability". In fact, familial history is reported in less than 10% of patients and not all RET mutations (identified in 20% of patients) are inherited but some of them are de novo.

2) 1 into 5000 is incidence and not prevalence

3) Classic or Rectosigmoid, Long, Total Colonic and Extended or Total Intestinal is the correct classification that divide the whole Hirschsprung population into 4 different severity subtypes. This is of utmost important particularly if we deal with stem cells treatment

4) again, I cannot agree with the statement "Mutations in RET, ZEB2, EDNRB, SOX10, L1CAM, PHOX2B, GDNF, NRTN, EDN3, ECE1, GFRA1, NRG1 were identified in HSCR patients, imparting up to 50% of the cases" as less than 25% of patients have mutations. To state that 50% have some mutation underlying the disease is incorrect

5) I would add rats to the list of animal models

Some more concerns are related to the central sections of the paper

6) The whole description of 2D and 3D stem cells based models is well written and clear. Even if Figure 2 is quite clear, I would add, if feasible, two more pictures or schemes clarifying most salient aspects of 2D and 3D models, respectively, leaving Figure 2 as a summary of pros and cons of the previously mentioned models

7) Again a picture clarifying the potentials of stem cells based treatments might be of use to the reader

8) The last section regarding future perspectives is unclearly identified to me. I do not see any difference in terms of perspectives amongst the various section of the paper. As a matter of facts, all models and treatments are still a measure of potentials and speculation. Based on these considerations, I would remove the title Future Perspectives and merely include the last two models as separate paragraphs numbered 2.3 and 2.4 or as separated paragraphs at the end of the manuscript, before conclusions section. This, in my mind, would avoid misinforming the reader on what is the present, the future and the ideas.

9) Even if I understand the limited value of conclusions in a review paper, I would have expected something more by the Authors to lead the readers into the future of stem cells models of Hirschsprung disease. For instance the limitation of sterile models, the importance of microbiome and the difficulties of re-creating the in vivo environment with its vascular supply (as stated in the organ-on-a-chip section) should be addressed to avoid semplification of such a complex but intriguing field of research.

10) Figure 3 is nice and is of great use to the reader to summarize future perspectives for stem cells based models

Round 2

Reviewer 2 Report

The Authors addressed most concerns. The paper now sounds